# Collagenase Activity of Bromelain Immobilized at Gold Nanoparticle Interfaces for Therapeutic Applications

**DOI:** 10.3390/pharmaceutics13081143

**Published:** 2021-07-27

**Authors:** Adrianne M. M. Brito, Vitor Oliveira, Marcelo Y. Icimoto, Iseli L. Nantes-Cardoso

**Affiliations:** 1Centro de Ciências Naturais e Humanas, Universidade Federal do ABC, Santo André CEP 09210-580, SP, Brazil; adriannemmb@gmail.com; 2Departamento de Biofísica, Escola Paulista de Medicina, Universidade Federal de São Paulo, São Paulo CEP 04039-032, SP, Brazil; vitor.oliveira@unifesp.br

**Keywords:** bromelain, gold nanoparticles, collagen, nanotechnology, cysteine protease

## Abstract

Bromelain (Bro) is a multiprotein complex extracted from the pineapple plant Ananas comosus, composed of at least eight cysteine proteases. Bro has a wide range of applications in medicine and industry, where the stability of its active proteases is always a major concern. The present study describes the improvement of stability and gain of specific activity in the enzymatic content of Bro immobilized on gold nanoparticles (GNPs). GNPs were synthesized in situ using Bro as the reducing and stabilizing agents and characterized by surface plasmon resonance and transmission electron microscopy. Consistent with the structural changes observed by circular dichroism analysis, the association with GNPs affected enzyme activity. The active Bro immobilized on GNPs (NanoBro) remained stable under storage and gained thermal stability consistent with a thermophilic enzyme. Proteolytic assays were performed on type I collagen membranes using fluorescence spectroscopy of O-phthaldialdehyde (OPA), changes in the membrane superficial structure, and topography by scanning electron microscopy, FTIR, and scanning laser confocal microscopy. Another characteristic of the NanoBro observed was the significant increase in susceptibility to the inhibitory effect of E-64, indicating a gain in cysteine protease activity. The higher stability and specific activity of NanoBro contributed to the broadening and improvement of Bro applications.

## 1. Introduction

### 1.1. Proteolytic Enzyme

Several enzymes have been purified partially or completely for therapeutic, biotechnological, and industrial applications, and proteolytic enzymes are extensively used for these purposes [1]. Proteolytic enzymes catalyze the hydrolysis of peptide bonds in a variety of biological processes. Proteases are classified according to the primary amino acid residue acting on catalysis and are thus divided into nine classes: cysteine, serine, aspartic, metallo, glutamic, threonine, and asparagine proteases, as well as mixed and unknown types [2]. Cysteine proteases (also called thiol proteases) are present in all living organisms and share a common catalytic mechanism involving a nucleophilic cysteine in the catalytic unit, which is composed of a triad of amino acids. The first isolated and characterized cysteine protease was papain obtained from papaya latex (*Carica papaya*). The enzyme from papaya became the model enzyme of the group known as the papain family. These enzymes have a structure with two main domains: an *N*-terminal helical domain and a β C-terminal leaf domain, containing the catalytic site with the characteristic cysteine amino acid residues, as well as histidine and asparagine [1,2,3]. Cysteine proteases are present in the pineapple-derived enzymatic complex known as bromelain (Bro).

### 1.2. Bromelain

Bro is a multiprotein complex extracted from pineapple (*Ananas comosus*), which was initially described as a pineapple-derived proteolytic aqueous extract. Bro stands out for at least eight cysteine proteases, and only a few of them have been sequenced and characterized with unique identifiers in the MEROPS (merops.sanger.ac.uk) and Uniprot systems, such as basic bromelain (or ananase, C01.005, EC 3.4.22.35), ananain (C01.026, EC 3.4. 22.31), and comosain (C01.027, with no defined EC number) [4,5,6].

### 1.3. Applications of Proteolytic Enzymes

The therapeutic use of plant proteolytic enzymes is old, with bromelain (pineapple), papain (papaya), and ficin (fig) the most commonly used. Although the advent of synthetic drugs has diminished these uses, the search for alternative treatments has aroused interest in natural and herbal medicines [7]. Currently, it is known that the fight against many diseases involves proteolytic activity of bioactive peptides and various proteins, such as signaling molecules in inflammatory, allergic, and tumor processes, bacterial membrane proteins, and virus capsids [8]. The applications and herbal uses of bromelain (Bro) in various areas, such as food, medicine, and cosmetics, have been studied for decades [9] and important therapeutic applications have been reported for Bro in studies demonstrating its in vitro and in vivo anti-inflammatory, anti-coagulant, anti-tumor, anti-metastatic, and immunogenicity properties [10]. Clinical studies have also demonstrated Bro’s effectiveness in the treatment of burns (debridement), although many of these mechanisms have not yet been fully elucidated [6].

In the industrial area, the proteolytic activity of bromelain is used in various sectors, such as food (production of hypoallergenic flours and protein hydrolysate, inhibition of fruit browning, production of alcoholic beverages, prevention of beer browning, and formation of protein precipitates in wine), the textile industry (minimizes break-in time, removes scales and impurities from wool and silk fibers and improves fiber dyeing properties), and dental/cosmetic (removal of stains, plaques, food debris, acne, wrinkles, and bruise reduction) [8,9,11]. Bro has been used in the food and cosmetic industries, and for environmental remediation.

Factors that hinder or limit some industrial uses of bromelain include denaturation and/or self-digestion in aqueous solutions, and the lack of a cost-effective purification process [12]. Like most proteins, enzymes can be denatured, and even small conformational changes can alter their activity, storage mode, storage time, and operational stability, which are limiting factors for several applications [13,14,15].

### 1.4. Importance of Nanotechnology for the Industrial Application of Proteases

The increase in protease stability is crucial for its application, which can be achieved by immobilization on nanoparticles (NPs) [16]. In addition, proteins can be used as reducing agents and templates for the synthesis of NPs, influencing their physicochemical characteristics. In particular, gold nanoparticles (GNPs) have high biocompatibility and unique optical, electronic, and catalytic properties [17,18]. Therefore, GNPs constitute an ideal platform for tuning the properties of bioactive proteins, such as enzymes. Currently, studies on the use of protein-associated nanoparticles have been increasing exponentially [7,19] as these systems are capable of improving stability and protecting them against enzymatic digestion denaturation, as well as increasing solubility and permeability, as well as reducing toxicity [20,21,22].

The present study describes the synthesis and characterization of gold nanoparticles (GNPs) produced using Bro as a reducing and stabilizing agent. The synthesis results in active Bro immobilized on GNPs (NanoBros) with a gain in thermal stability and aging resistance.

## 2. Materials and Methods

Reagents—tetrachloroauric salt trihydrate (HAuCl_4_.3H_2_O), HEPES, sodium phosphate dibasic anhydrous, bromelain from pineapple stem powder ≥ 3 U/mg, papain from *Carica papaya* powder ≥3 U/mg, protease inhibitor E-64, (*N*-[*N*-(L-3-transcarboxyirane-2-carbonyl)-L-leucyl]-agmatine), and o-phthalaldehyde (OPA) were acquired from Sigma-Aldrich Chemical Co. (St. Louis, MO, USA). Organic matrix of type I collagen purified, polymerized, fibrillar, Hemostatic SURGIDRY (membrane) produced and donated by TechnoDry Liofilizados Médicos Ltd.a. (Belo Horizonte, MG, Brazil).

Spectroscopy UV-visible—Absorbance measurements were carried out using a UV-visible photodiode array spectrophotometer MultiSpec-1501 (Shimadzu Co., Kyoto, Japan). For the measurements, the samples were loaded in quartz cuvettes with optical paths of 1.0, 0.5, and 0.1 cm, according to the requirement. The synthesis temperature of the GNPs was controlled using a thermal bath coupled to a spectrophotometer.

Circular dichroism (CD) spectroscopy was used to analyze the structural changes in the secondary structure of bromelain when associated with GNPs in relation to aging and temperature variation. The CD measurements were carried out in a J-815 circular dichroism spectrometer equipped with a Peltier single cell holder (Jasco International Co., Ltd., Tokyo, Japan) using quartz cuvettes with a 10 or 5 mm optical path for measurements of bromelain solution (0.05 Â mg/mL) and bromelain associated with gold nanoparticles (<0.05 Â mg/mL); bandwidth, 1.0 nm; scanning speed, 50 nm/min and accumulations, 10.0.

Fluorescence—Fluorescence spectra were obtained using a Cary Eclipse fluorescence spectrophotometer (Varian) and quartz cuvettes, with 1.0 cm optical path. The emission and excitation wavelengths used were specified in the results.

Confocal microscopy—Confocal images were obtained using a Zeiss microscope (LSM 510-Meta, Oberkochen, Germany) with a laser of 633 nm. The surface roughness and profiles were determined in a representative region of 2500 μm^2^ at a magnification of 500×. The measurements were performed at three points in the selected area and at a depth of 50 μm.

Transmission electron microscopy—The images were obtained using TEM Tecnai G2 Spirit BioTWIN, at 80 Kv, in the LME/IB of UNICAMP. The solution was allowed to evaporate before the microscopy was performed. The average size of the AuNPs was determined using Image J software [23].

Fourier transform infrared (FTIR) spectra were obtained using a Spectrum Two FTIR spectrophotometer with a Universal ATR sampling accessory (PerkinElmer, Madson, WI, USA.) using crystal diamond with 32 scans from 700 to 4000 cm^−1^ and a resolution of 4 cm^−1^.

All syntheses were performed in a laminar flow cabinet using small lab glass vials with a black screw cap. The glassware used for synthesis was previously cleaned to avoid undesired nucleation aggregation of colloidal gold solutions [24]. The addition of reagents was performed as follows: Milli-Q deionized water, 0.2 mM sodium phosphate buffer plus HEPES, pH 10, 100 μL of protein solution per mL; tetrachloroauric acid solution (400 μM). GNPs were synthesized at 40 °C for 2 h in a vacuum dry bath and allowed to stabilize at room temperature. After 24 h of synthesis, the GNP colloidal suspensions were centrifuged and resuspended in 0.2 mM sodium phosphate buffer plus HEPES (pH 7.0) and stored at 4 °C. Several syntheses were performed and characterized using UV-vis spectroscopy to guarantee reproducibility.

Collagen membrane digestion—Samples of type I collagen membranes were subjected to enzymatic degradation by bromelain in the free form and conjugated to GNPS. As controls, papain-free or associated with GNPs (synthesized under the same conditions as BroGNP) or E-64, a cysteine protease inhibitor, were added to the enzymes. Phosphate-buffered saline (PBS, 10×) was used as a negative control. In this assay, 1 mg (± 0.1) rectangles from the same collagen membrane were cut, weighed, and subjected to proteolysis in Eppendorf^®^ microtubes (Hamburg, Germany) containing 300 μL of the enzyme-free solution or associated with GNPs, for 72 h, except for the papain group. In the papain group, the process was carried out in duplicate, and in one, the membrane was removed after 4 h of reaction, and in the other, it was observed that the total degradation of the membrane occurred after 12 h of reaction. After the indicated period, the membrane and the supernatant solution were separated and prepared for analysis by UV-Vis spectroscopy and fluorescence spectroscopy after reaction with OPA, FTIR, SEM, and CLSM. For FTIR, CLSM, and SEM analyses, the membrane samples were dried by evaporation at room temperature in a container with silica. For the SEM analysis, the samples were covered with gold. The working solution had a final concentration of 1 mg/mL OPA in 0.1 M PBS pH 7.4 with 2.4% ethanol and 0.2% β-mercaptoethanol. Fluorescence was obtained by excitation at 340 nm and emission at 455 nm. The fluorescence measurement was performed 5 min after the addition of OPA to a volume of 100 µL of the solution resulting from the enzymatic activity.

## 3. Results and Discussions

NanoBro was synthesized in situ using Bro as a reducer agent and a biotemplate for the growth of gold nanocrystals, and the nanostructures were characterized and applied for collagenolytic activity.

### 3.1. Synthesis and Characterization of the NanoBro

Figure 1 shows the UV-vis spectral changes observed during NanoBro synthesis started by mixing HEPES-buffered Bro solution (green line) with HAuCl_4_ (orange line). Thirty seconds after mixing Bro and HAuCl_4_ solutions, a decrease in the gold salt band at 290 nm was observed, followed by a progressive increase in the surface plasmon resonance (SPR) in the spectral region of 500–600 nm. During GNP formation, the SPR band presented a progressive blueshift, probably because of the deaggregation promoted by protein capping [25]. In Figure 1, the inset shows a snapshot of the HEPES-buffered Bro transparent solution at pH 7.0, yellow HAuC aqueous solution, and red-wine GNP suspension. In the early steps of the synthesis, the decrease in absorbance at 290 nm was attributed to the consumption of gold ions by the reduction process, and the increase in absorbance at 290 nm coincident with the SPR appearance and growth is consistent with the formation of gold nanocrystals functionalized by the Bro molecules.

The effect of HEPES on the synthesis of NanoBro was investigated. Figure 2a (wine lines) shows the UV-Vis spectra of gold colloidal suspensions obtained with HEPES-buffered Bro (NanoBro, line 1), with HEPES and sodium phosphate alone [26] (line 2) and with Bro aqueous solution (line 3), and the respective snapshots show the features of the colloidal suspensions. GNPs produced with HEPES without Bro (line 2) exhibited spectral features indicative of polydispersivity and anisotropy, whereas Bro alone produced GNPs with low yield and without colloidal stability (line 3). The green line 5 shows that sedimentation of NanoBro by centrifugation at rpm resulted in a supernatant in which the spectrum peaking at 280 nm, characteristic of the contribution of aromatic amino acids, is absent. Therefore, a larger fraction of the proteins was associated with the GNPs, resulting in efficient capping of GNPs by the Bro enzymes. NanoBro are spherical nanoparticles with a mean diameter of 5.0 ± 1.3 nm, as shown in Figure 2b,c. In addition to the low colloidal stability leading to aggregation and precipitation, GNPs produced with Bro alone (line 3) exhibited a slow rate of synthesis that required more than 24 h for completion. The low colloidal content of GNPs produced with Bro alone could be due to the extensive oxidation of the thiol and amino groups of cysteine and lysine residues, impairing protein binding to the gold surface and promoting crosslinking between protein chains. The presence of HEPES buffer contributes to rapid nucleation (nanoclusters), preserving the enzyme structure against extensive oxidation and consequently contributing to efficient nanoparticle capping and colloidal stability. In this regard, the three types of gold colloidal suspensions were subjected to centrifugation for NP precipitation. However, only NanoBro synthesized with HEPES assistance could be resuspended and presented an absorbance spectrum similar to that obtained before centrifugation. The sample synthesized only with HEPES buffer lost its reddish coloration, which could be attributed to gold oxidation or extensive GNP aggregation that shifted the spectra to the infrared spectral region. The sample synthesized only with Bro remained precipitated, consistent with the fact that in the nucleation, the reducing groups (thiols and lysine) were oxidized, impairing the efficient anchorage of the enzyme on GNPs. Taken together, these data suggest that HEPES is a crucial reducer agent for gold nanocluster nucleation and Bro as a biotemplate for nanocrystal growth and stabilization.

### 3.2. Structural Bro Stabilization in NanoBro

Figure 3a shows the circular dichroism (CD) spectrum of Bro overlapped by the typical CD spectra of three pure secondary structures: α-helix, β-sheets, and random coil produced with the datasheet of [27]. The bromelain spectrum is a composite spectrum with α-helix and β-sheet spectral contributions, which is consistent with an α + β protein class, as well as other cysteine proteases [28]. Enzymes of this family have been reported to contain 23% α-helix, 5% parallel β-sheet, 18% antiparallel β-sheet, 28% turns, and remaining percentage of other (unidentified or random) secondary structures [28]. Spectral decomposition of the Bro CD spectrum by BeStSel (Online tool http://bestsel.elte.hu/index.php, accessed on 8 April 2018) [29] showed the contributions of 24% α-helix, 0% parallel β-sheet, 24.2% antiparallel β-sheet, 11.9% turns, and 39.3% random coil.

Bro solution stored at room temperature for two weeks (Figure 3b) presented changes in the secondary structure, particularly a significant loss of α-helix content from 24.5% to 5.2% with an increase in the anti-parallel β-sheet and random coil content, as depicted in Table 1. NanoBro exhibited a similar secondary structure content, which remained unchanged after two weeks of storage at room temperature (Table 1). Therefore, binding to GNPs increases the stability of the enzyme during storage. In the analysis of these CD spectra, it should be considered that bromelain corresponds to a set of enzymes and the spectrum represents the sum of the spectral contributions of different protein fractions, and there is the possibility that one fraction was not totally or partially retained on the GNPs, changing the percentages of secondary structures.

Considering that Bro is composed of a mix of proteolytic enzymes, the improvement of the time-dependent stability could result from the immobilization that hinders the access of one enzyme to the active site of another, the exclusion of one enzyme fraction mainly responsible for Bro self-degradation, and even the protection of some enzyme groups from the oxidative degradation in air atmosphere. The mechanisms underlying the time-dependent stability gain of NanoBro warrant further investigation. The most noteworthy result observed by the Far-UV-CD analysis (Figure 3c,d) was that NanoBro gained thermal stability, becoming practically thermophilic. Figure 3c,d show the CD spectra of Bro in solution and NanoBro after incubation at different temperatures in the range of 20–80 °C. Consistent with literature data [9,12], Figure 3c shows that Bro in solution exhibits temperature-dependent denaturation with a turning point at 60 °C, as demonstrated in the inset of Figure 3c. In the inset of Figure 3c,d, the maximal positive (195 nm) and negative (222 nm) intensities of the spectral contribution of α-helix content were used to show the effect of temperature on the secondary structure of the enzyme. NanoBro exhibited CD spectra with similar features over the investigated temperature range (Figure 3d). Enzymatic activity is very likely to remain stable at high temperatures (>60 °C) when immobilized on GNPs [30]. This thermal stability could be suitable for industrial applications where a faster reaction rate is desirable as it can be used at a much higher temperature, greatly accelerating the reaction rate without losing activity and increasing efficiency. A possible application would be in effluent or related cleaning.

### 3.3. Comparative Study of Enzymatic Activity on a Collagen Membrane

Considering that collagenolysis is important for biomedical and industrial applications, the collagenolytic activity of NanoBro was investigated [31]. For this purpose, type I collagen membrane samples were subjected to enzymatic degradation using a Bro solution and NanoBro. In the collagenolytic assay, the membrane was immersed in a saline solution without (negative control) and with Bro and NanoBro. The collagenolytic activity was evaluated by UV-visible spectroscopy, OPA fluorescence assay, and FTIR of the supernatant, as well as analysis of the membrane by FTIR, MEV, and CLSM. Despite the fact that bromelain has a relatively low inhibition by E-64 [32], in some assays, this cysteine protease inhibitor was added to the incubations and compared with the activity of papain, which is very susceptible to inhibition by E-64. In these assays, papain was used in the solution and immobilized on GNPs.

#### 3.3.1. UV-Vis Absorption Spectroscopy Analysis of the Supernatant Solution

Incubation of the type I collagen membrane at 25 °C for 22 h resulted in progressive decoloration of the colloidal suspension with a concomitant change in the membrane color from opaque white to red color, indicating impregnation of NanoBro in the membrane structure (inset of Figure 4). The time-dependent color changes of the colloidal suspension were accompanied by spectral changes in the UV region of the spectrum with the appearance of a band peaking at 280 nm, which is consistent with the increased concentrations of peptides (Figure 4).

#### 3.3.2. Fluorescence Analysis of the Supernatant after Derivatization with Ortho-Phthalaldehyde (OPA)

The collagenolytic activity of Bro and NanoBro indicates the production of peptides that can be detected by the reaction with o-phthalaldehyde (OPA), which reacts with primary amines of peptides to allow fluorescent detection and quantitation. OPA assays were carried out with solutions in which the collagen membranes were subjected to the proteolytic activity of Bro, NanoBro, and the supernatant of NanoBro synthesis. The membranes were incubated with PBS in the absence of enzymes. The use of the supernatant of NanoBro synthesis had the objective of detecting reminiscent enzymatic activity from Bro fractions that did not bind to GNPs. Each condition was tested in the absence and presence of E-64 to determine the contribution of cysteine protease activity to collagen hydrolysis. E-64 is selective for cysteine proteases and acts irreversibly by covalent binding with the reactive thiol group of the active site. Figure 5a shows a schematic illustration of the mechanism of the OPA assay for the relative quantification of peptides produced under each experimental condition. Figure 5b–d shows the relative fluorescence intensity obtained in the absence (solid line) and presence (dashed line) of E-64 for the enzymatic activity of Bro, NanoBro, and the supernatant of NanoBro synthesis, respectively. The fluorescence of the control solution is indicated by the red dashed lines. The fluorescence spectra obtained under each condition are presented as a bar graph showing the corresponding percentage of inhibition promoted E-64 and the difference (Δ) between the total enzymatic activity and the reminiscent activity in the presence of E-64, which corresponds to the percentage of cysteine protease activity of Bro, NanoBro, and the Bro fractions that remained free in solution. Considering that the non-enzymatic membrane fragmentation resulted in the same fluorescence intensity obtained with NanoBro and E-64, only cysteine protease activity was present in the enzymatic complex bound to GNPs. The lowest percentage of inhibition by E-64 was observed in the supernatant of NanoBro synthesis. Therefore, the enzymatic activity of NanoBro is exclusively a cysteine protease that could result from the selective binding of cysteine proteases by GNPs or inhibition of the bound enzymes that are not cysteine proteases. The preferential retention of cysteine protease fractions by GNPs is less probable because E-64 efficiently inhibits only ananain and comosain, which constitute only 6% of the total protein in bromelain. Thus, GNPs probably favor the activity of ananain and comosain, and this issue deserves further investigation.

#### 3.3.3. Confocal Laser Scanning Microscopy (CLSM) Image-Based Analysis

Changes in the collagen membrane surface promoted by NanoBro were characterized using CLSM [28,33,34]. CLSM was employed to construct the roughness profile of the collagen membrane subjected to collagenolytic activity of Bro and NanoBro compared with papain. Two scanning modes were used: optical scanning to verify membrane surface changes (Appendix A) and vertical laser scanning to show the differences in topography (Figure 6a–d). The analysis of collagen membrane topography showed that NanoBro, similar to papain, promoted the leaching and smoothing of the membrane, and the leaching was particularly decreased by E-64. The collagenolytic activity of NanoBro was also evaluated by FTIR spectroscopy of the membrane and supernatants.

#### 3.3.4. Analysis of Chemical and Structural Changes of Collagen by ATR-FTIR

In the present study, ATR-FTIR was used to analyze the surface layer of the sample [35,36]. Figure 7 shows the ATR-FTIR spectra obtained from the collagen membrane samples (thick solid lines) and the supernatant solution resulting from the enzymatic hydrolysis (thin solid lines), in the spectral range of 1700 to 800 cm^−1^, which are attributed to the vibrations of specific functional groups of amino acids, proteins, and peptide chains. In the collagen membrane spectra, it was possible to identify the absorption bands between 1700 and 1300 cm^−1^ relative to amide I (~1650 cm^−1^), amide II (~1550 cm^−1^), and amide III bands (1400–1200 cm^−1^) [37]. No changes were observed in the spectral regions of amides I, II, and III of the membranes incubated with buffer or enzymes. This was expected because the proteolytic activity promotes leaching of the membrane layers but does not change the secondary structure of the reminiscent layers. The presence of a triple helical structure can be determined using the intensity ratio of amide III peaks at 1233/1450 cm^−1^ [37]. The presence of a triple helix was indicated by an absorption ratio above 1.0. The collagen membrane used here exhibited slight variations in the maximum at the 1200 cm^−1^ region (see indications in Figure 7), and the maximal value was used for the ratio calculation. The collagen membrane preserved the 1237–9/1450 cm^−1^ ratio >1.0 after treatment with Bro and NanoBro whereas the ratio becomes slightly < 1.0 after the exposure to papain. Consistent with the specific cleavage of collagen chains promoted by papain (orange line), Bro, and NanoBro, significant changes in the spectral regions corresponding to the side chains of some amino acids were observed in the membrane FTIR spectra, indicating a change in the microenvironment of these groups. The cleavage of collagen chains can modify the amino acid side chain microenvironment, such as the decrease in the spectral contribution around 1063 cm^−1^, which occurs only in Bro-and NanoBro digested samples. This band is attributed to the stretching of the C-C skeleton in a random conformation [38]. The presence of collagen chain fragments in the supernatant as a result of the digestion process is evident by the drastic change in the intensity ratio of the collagen-related helix bands (1700–1200 cm^−1^) with those related to the free amino acid side chain peptides (1200–800 cm^−1^). The peak absorption bands in the 853 cm^−1^ spectral region are related to the vibrations of the collagen-specific amino acids proline and hydroxyproline [38,39]. Thus, as an example, in the FTIR spectra of the membranes, the 1642/865 cm^−1^ absorbance ratio has a mean value of 1.15, while in the FTIR spectra of the supernatants, the ratio has a mean value of 0.89. However, in the spectrum of the supernatant containing the products of papain proteolysis, a higher contribution of amide I, II, and III bands was evident, and the 1642/865 cm^−1^ ratio in this spectrum was 0.92. This result suggests that free papain activity produces larger collagen fragments that retain some secondary structure. These results reinforce that Bro has collagenase activity preserved in NanoBro.

## 4. Conclusions

Characterization of the in situ synthesis of GNPs using bromelain and its effects on the structure and activity of the enzyme were demonstrated. As for the synthesis protocol, it was observed that bromelain can be used as a reducing and stabilizing agent for the synthesis of GNPs, but the stabilizing capacity is more efficient when the HEPES buffer is present, possibly because it acts as a nucleating agent and thus preserves the reducing groups of the enzyme, especially the thiols that are mainly responsible for coordination with the surface.

Studies related to the structure of the enzyme associated with nanoparticles have shown that this association affects the structure and stability of the enzyme. After synthesis, bromelain was found to be a stabilizing agent for GNPs. This association promoted a gain in structural stability as the temperature increased, so that the enzyme became thermophilic and gained more structural stability during storage. Associated with the GNPs, Bro presented increased alpha-helix content possibly by the acid microenvironment at the nanoparticle interface.

Consistent with the observed structural changes, the association of Bro with nanoparticles affected the activity of the enzyme. In collagen membranes, it was verified that the interaction of NanoBro with these membranes promoted their partial digestion, but in a differentiated way concerning the free enzyme and papain. Another differential observed for NanoBro was the significant increase in sensitivity to inhibition by E-64, that is, gain of specific cysteine protease activity. These differences could be attributed to both the binding of specific protein fractions to the GNPs and structural changes in the protein content linked to the nanostructures. As the protein dosage measurements of the supernatant of the samples after the synthesis of GNPs with Bro showed a very low concentration of unbound proteins, the stability changes and the specificity were attributed to the structural alterations.

Thus, the numerous applications of bromelain and the factors that limit its use justify the investigation of its enzymatic activity associated with GNPs. Particularly with regard to large-scale applications, the effects of structural stabilization and increased specificity provided by Bro in association with GNPs are auspicious, contributing significantly to a better understanding of the activities that make up Bro extract.

## Figures and Tables

**Figure 1 pharmaceutics-13-01143-f001:**
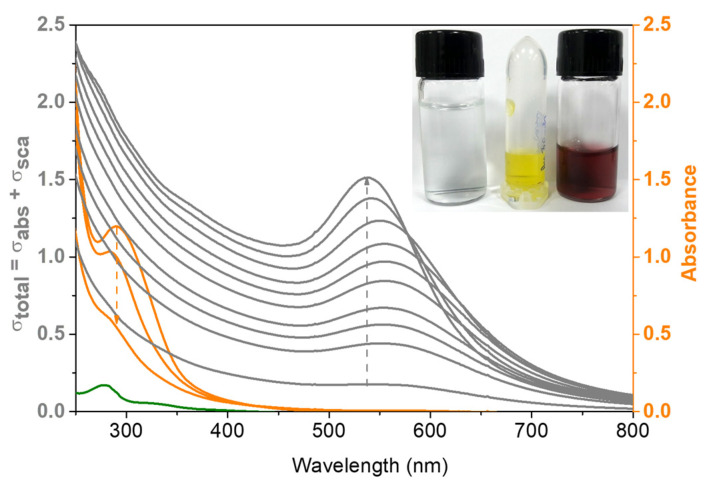
UV-Vis spectra acquired during the time-course of the NanoBro synthesis. Green line shows the HEPES-buffered Bro solution. Orange lines show the spectral changes of HAuCl_4_ in subsequent times after addition to Bro solution as indicated by the arrow. Gray lines show the increase of absorbance and turbidity resulted from NanoBro formation. The inset shows a snapshot of the HEPES-buffered Bro transparent solution, the yellow HAuCl_4_ aqueous solution and the red-wine GNP suspension. The synthesis was carried out at pH 7.0 at the temperature of 40 °C and the spectra were obtained in a quartz cuvette with 1 cm of optical path.

**Figure 2 pharmaceutics-13-01143-f002:**
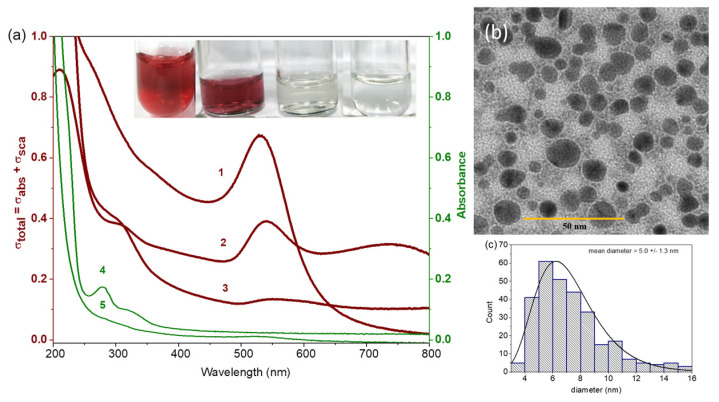
Characterization of nanoparticles. (**a**) UV-Vis spectra of NanoBro colloidal suspension produced with HEPES-buffered Bro (line 1), with HEPES and sodium phosphate (line 2), with aqueous solution of Bro (line 3). Green lines 4 and 5 correspond, respectively, to the spectra of HEPES-buffered Bro (0.2 mg/mL) and the supernatant of NanoBro synthesis (line 1) obtained by GNPs precipitation by centrifugation. The inset shows, from left to right snapshots of solutions 1–4. Spectra were obtained 4 days after the synthesis; (**b**) transmission electron microscopy (TEM) of NanoBro was obtained with magnification of 100,000× and energy of 80 kW; (**c**) histogram of NanoBro size distribution fitted as lognormal curve by Origin tool calculated a mean diameter of of 5.0 ± 1.3 nm.

**Figure 3 pharmaceutics-13-01143-f003:**
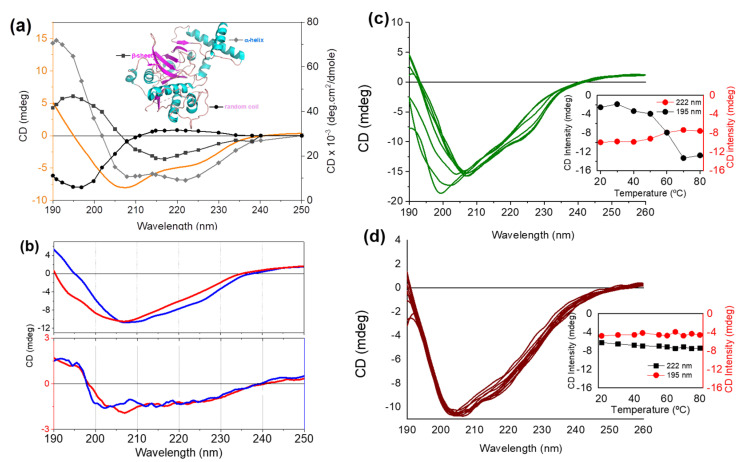
Structural characterization and stability of Bro and NanoBro by circular dichroism (CD) spectroscopy. (**a**) HEPES buffered Bro solution spectrum overlapped with CD spectra of three pure secondary structures, α-helix, β-sheet, and random coils obtained from [27]. The inset shows the PDB structure of a recombinant bromelain (code 6U7D) to illustrate the presence of these structures in the enzyme. (**b**) Effect of storage time on the Bro (upper panel) NanoBro (lower panel) secondary structures. Blue and red lines correspond, respectively, to the CD spectra obtained immediately and two weeks after storage. (**c**,**d**) Effect of temperature in the range of 20–80 °C on the CD spectra of Bro solution and NanoBro, respectively. The native protein in aqueous solution 0.2 mg/mL has pH 5.2 and colloidal suspension of NanoBro, pH 6.5.

**Figure 4 pharmaceutics-13-01143-f004:**
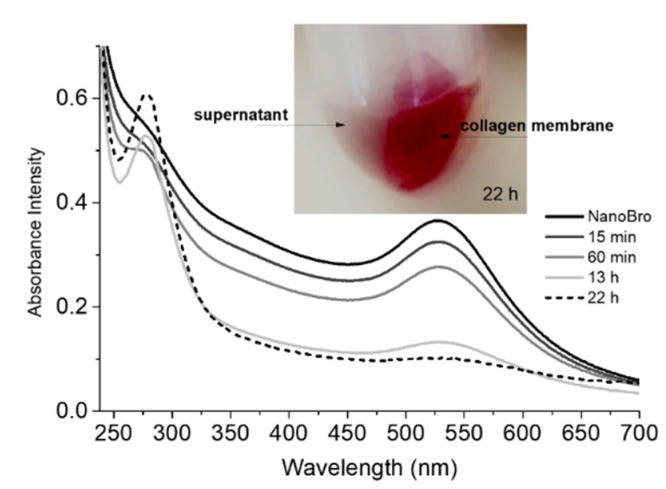
Binding affinity and proteolytic activity of NanoBro on Type I collagen membranes. UV-Vis spectra of progressive bleaching of NanoBro colloidal suspension during 22 h of incubation with Type I collagen membrane with concomitant increasing of the absorbance of peptide chains. NanoBro colloidal suspension (black line) was incubated with the collagen membrane and the spectra of the supernatant was collected and analyzed by UV-vis spectroscopy after 15 min (dark grey line), 1 h (grey line), 13 h (light grey line), and 22 h (black dotted line). Peak at 280 nm is a contribution of peptide fragments leached from the membrane and absorbance peaking at 520 nm is the contribution of NanoBro SPR band. The inset shows collagen membrane red colored by the adsorption of NanoBros after 22 h of incubation.

**Figure 5 pharmaceutics-13-01143-f005:**
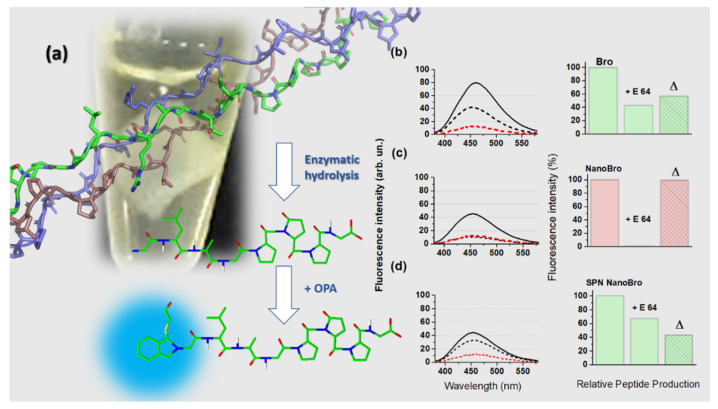
Relative quantification collagen membrane proteolysis promoted by Bro, NanoBro and reminiscent Bro not bound to GNP, by the detection of peptides in the supernatant using OPA assay. The supernatant was treated with OPA and analyzed by fluorescence emission spectroscopy with excitation at 340 nm and emission spectra acquisition in the 380–580 spectral range. (**a**) Schematic illustration of the enzymatic activity of NanoBro on collagen followed by the reaction with OPA. A recombinant human collagen from PDB (1BKV) was used to illustrate a collagen triple helix and a presumed peptide fragment from the collagen sequence was designed using Avogadro software with its structure before and after reaction with OPA; (**b**–**d**) show the respective fluorescence spectra of OPA-derativized peptides obtained from the supernatant of collagen membrane incubated with Bro, NanoBro and reminiscent Bro in the absence (**solid lines**) and presence (**dashed lines**) of E-64. At the right side of the fluorescence spectra, it is shown the corresponding bar graphs (green corresponds to free enzyme and red to NanoBro) indicating the percentage of E-64 inhibition and the contribution of cysteine-protease activity of each sample (Δ). A total of 100 μL of enzyme solution was used in all groups.

**Figure 6 pharmaceutics-13-01143-f006:**
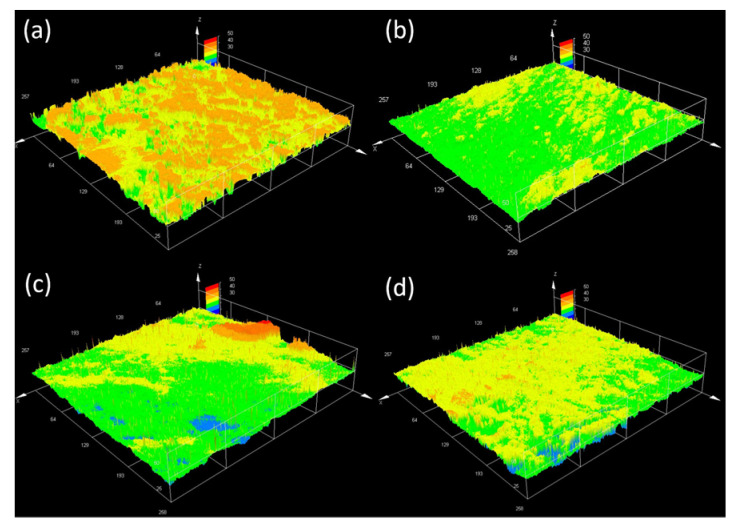
Confocal laser scanning microscopy (CLSM) showing the topography of collagen membrane submitted to the enzymatic activity of NanoBro and papain. (**a**) In a negative control corresponding to the membrane incubated with HEPES buffer; (**b**) positive control corresponding to collagen membrane incubated with papain; (**c**,**d**) collagen membrane incubated with NanoBro in the absence and in the presence of E-64. The images were obtained by vertical scanning of 50 µM depth in horizontal planes between the surface and the maximum depth adopted.

**Figure 7 pharmaceutics-13-01143-f007:**
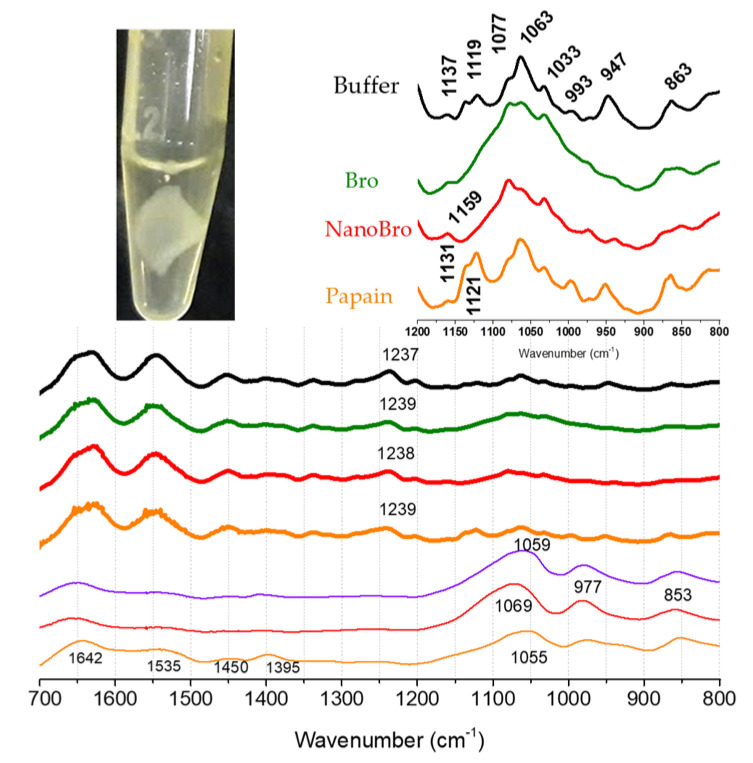
FTIR spectra of collagen membranes (tick solid lines) and supernatant solution (thin solid lines) after enzymatic activity of papain (orange lines), NanoBro (red lines) and Bro (green lines). Membrane spectrum incubated with buffer alone is represented by thick solid black line.

**Table 1 pharmaceutics-13-01143-t001:** Summary of the estimate of the percentage of secondary structures found in Bromelain under various experimental conditions according to data obtained by Circular Dichroism and analyzed by BeStSel.

	Fresh Bro	Stored Bro	Fresh NanoBro	Stored NanoBro
Helix (total)	24.5	5.2	14.8	22.4
Antiparallel (total)	24.2	35.8	28.9	25.0
Turn	11.9	11.4	12.8	11.6
Others	39.3	47.6	43.5	41.0
The CD units were converted to delta epsilon in the range of 190–250 nm.				

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
