# Peer review of "Collagenase Activity of Bromelain Immobilized at Gold Nanoparticle Interfaces for Therapeutic Applications"

_pharmaceutics, 2021, doi:10.3390/pharmaceutics13081143_

Round 1

Reviewer 1 Report

The authors provided a synthesis method to immobilize Bromelain (Bro) on gold nano particles (GNPs) and applied various characterization methods such as CD/ FTIR/ enzyme activity FL test/ Confocal to find that Bro-GNP could maintain activity and stability of Bro under storage.

  1. Figure 3 has 2 (c)s, please correct the labeling.
  2. The caption for Figure 3 "Blue and red lines correspond to the CD spectra obtained immediately and two weeks after NanoBro synthesis, respectively (c) and (d) Effect of temperature in the range of 20 to 80 C on the CD spectra of Bro solution (c) and NanoBro(d). " is confusing while mentioned (c) and (d) in the sentence describing figure 3b.
  3. The y-axis of insets in figure 3c and figure 3d should be CD intensity instead of 222 nm or 195 nm. 195 nm and 222 nm should be legends.
  4. Line 232-233: Bro solution stored at 4°C for only one week (Figure 3b) presented a significant loss of α-helix content to 5.2% with a significant increase of anti-parallel β-sheet to 35.8% and random coil to 47.6% without significant changes in the turn content (11,4%). The sentence requires rephrase. Also, there is no Bro solution sample in the figure 3b.
  5. Instead of describing the de-convolution structure change by words, I suggest the authors put together a table to show how the α-helix/anti-parallel β-sheet/random coil/turn content change in both bro and nanobro solution along with time.
  6. Figure 3d data can't prove that nanobro maintains enzyme structure. Since most of enzymes are attached on the GNP, it will be hard to detect structure change of enzyme while temperature increases in the presence of GNP.
  7. Please show legends in the Figure 4
  8. Figure 5, the authors should also provide the FL intensity in the absence of any bro enzyme and with E64 only as control.

Author Response

We thank Reviewer 1 for the review of the manuscript.

Marcelo Y. Icimoto and Iseli L. Nantes-Cardoso

Reviewer 2 Report

The authors describe the use of bromelain to synthesise gold nanoclusters to produce protein-nanocluster conjugates.  These conjugates display enhanced stability and activity, which the authors characterise by various spectroscopic methods and enzyme assays.

I found the work interesting and generally well written and presented.  I have a few minor queries and suggestions that should be addressed.

The introduction is quite dense in its present form and should be subdivided into shorter paragraphs.

In the methods section more detail should be included for fluorescence spectroscopy (for example, wavelengths used), and how SEM and TEM data were measured (fro example, magnifications etc)

The nanocluster sysnthesis described on page 3 (lines 119-129) mentions phospahte buffer with HEPES at pH 10.  Is this correct neither HEPES or phosphate have much buffereing capacity at a pH this high.  Were other sulfonic acid buffers containing buffers tested for example CHES which can buffer in this range?  How stably was the pH maintained during the reaction?  

In figure 3 there are two panels labelled c, also the insets in what I believe to the correct panels c and d showing the change in specific CD features with temperature should be mentioned in the legend.  Furthermore the discussion relating to the changes in CD features with temperature (page 8) should be expanded slightly to clarify what the changes in absorbance at 195 and 222 nm mean in terms of protein structure.

Adding times to UV spectra would help distinguish the spectra.

I'm not sure I understand the discussion regarding increased cysteine protease activity of NanoBro relative to Bro.  It seems like the GNPs are binding cysteine proteases from the solution resulting in increased activity, is this correct?  Could this be a relative concentration effect?  The sentence (lines 328-331) ...it is more probable that GNPs favour cysteine protease fractions of Bro rather than selective binding of cysteine protease fractions to GNPs... is causing my confusion and the meaning should be clarified.

Figure 7 legend - replace tick with thick

Author Response

We thank reviewer 2 for the review of the manuscript.

Marcelo Y. Icimoto and Iseli L. Nantes-Cardoso

Round 2

Reviewer 1 Report

Thanks for addressing all the comments. I recommend to accept the revised article.